# Laser Surface Modification of TC21 (α/β) Titanium Alloy Using a Direct Energy Deposition (DED) Process

**DOI:** 10.3390/mi12070739

**Published:** 2021-06-24

**Authors:** Ahmed Magdi Elshazli, Ramadan N. Elshaer, Abdel Hamid Ahmed Hussein, Samar Reda Al-Sayed

**Affiliations:** 1Department of Engineering Applications of Laser, National Institute of Laser Enhanced Sciences (NILES), Cairo University, Giza 12611, Egypt; eg.engshazly@yahoo.com; 2Department of Mechanical Engineering, Tabbin Institute for Metallurgical Studies (TIMS), Cairo-Egypt, Helwan 11731, Egypt; ramadan_elshaer@yahoo.com; 3Department of Metallurgy, Faculty of Engineering, Cairo University, Giza 12611, Egypt; aahussein41@yahoo.com

**Keywords:** TC21 titanium alloy, direct energy deposition (DED), microstructure, stellite-6, tungsten carbides particles (WC), microhardness, wear resistance

## Abstract

The TC21 alloy (Ti-6Al-3Mo-1.9Nb-2.2Sn-2.2Zr-1.5Cr) is considered a new titanium alloy that replaced the commercial Ti-6Al-4V alloy in aerospace applications due to its higher operating temperatures. Recently, direct energy deposition was usually applied to enhance the hardness, tribological properties, and corrosion resistance for many alloys. Consequently, this study was performed by utilizing direct energy deposition (DED) on TC21 (α/β) titanium alloy to improve their mechanical properties by depositing a mixture powder of stellite-6 (Co-based alloy) and tungsten carbides particles (WC). Different WC percentages were applied to the surfaces of TC21 using a 4 kW continuous-wave fiber-coupled diode laser at a constant powder feeding rate. This study aimed to obtain a uniform distribution of hard surfaces containing undissolved WC particles that were dispersed in a Co-based alloy matrix to enhance the wear resistance of such alloys. Scanning electron microscopy, energy dispersive X-ray analysis (EDAX), and X-ray diffractometry (XRD) were used to characterize the deposited layers. New constituents and intermetallic compounds were found in the deposited layers. The microhardness was measured for all deposited layers and wear resistance was evaluated at room temperature using a dry sliding ball during a disk abrasion test. The results showed that the microstructure of the deposited layer consisted of a hypereutectic structure and undissolved tungsten carbide dispersed in the matrix of the Co-based alloy that depended on the WC weight fraction. The microhardness values increased with increasing WC weight fraction in the deposited powder by more than threefold as compared with the as-cast samples. A notable enhancement of wear resistance of the deposited layers was thus achieved.

## 1. Introduction

Nowadays, titanium, and particularly its alloys, have received more attention for a wide range of applications in many fields, such as military, medical, and civil transportation, especially (α + β) titanium alloys [1,2,3]. They are widely used in advanced aerospace applications, aero-engines, and chemical industries due to their unique merits of low density, high strength-to-weight ratio, heat resistance, and corrosion resistance [4,5]. All these characteristics make them the most favorable material choices for certain applications, such as airplanes, making up to 30−50% weight of the total structure [6,7]. Recently, the TC21 titanium alloy (Ti-6Al-2Sn-2Zr-3Mo-1Cr-2Nb-Si (wt.%)) is considered a new version in aerospace applications that replaced the commercial Ti-6Al-4V grade five titanium alloys. It belongs to the family of (α + β) titanium alloys with high strength, toughness, and damage-tolerance properties [8]. With the use of the TC21 alloy in aircraft components, such as landing gear or the flap track, there will be a significant advantage in terms of a weight reduction for such aircraft [9]. However, such components are subjected to many stresses and friction wear during operation. Accordingly, the applications of such alloys under severe wear conditions are highly restricted due to their low hardness and poor tribological properties. Normally, their mechanical properties are strongly associated with their microstructures, which can be developed during various modes of heat treatment [10].

Laser surface engineering is a novel approach for the surface modification of metals that are aimed at enhancing their performance when subjected to severe environmental and industrial conditions [11]. Laser processing, which is a newly developed technique, offers many attractive advantages over conventional coating techniques [12]. The direct energy deposition (DED) process is a new rapid manufacturing technology that can build fully dense metal components directly from a selected powder [13]. It involves using the concentrated energy of lasers with material interaction to re-engineer a metallic surface. In addition, DED offers some advantages over conventionally used methods, including a high deposition rate, low dilution of the substrate, and low distortion [14].

Several studies have investigated the direct energy deposition process for different titanium alloys, especially the commercially pure titanium and the Ti-6Al-4V titanium alloys [15,16]. For example, Sampedro et al. [17] investigated the defect-free coatings of Ti-6Al-4V and the mixture of (Ti-6Al-4V + TiC) on grade 2 titanium and Ti–6Al–4V alloys, respectively, which were created using the direct energy deposition process. This study aimed to improve the mechanical properties of such alloys. The results reported a twofold hardness improvement compared with the substrate. The hardness level depended on the weight percent of TiC in the Ti-6Al-4V powder. Additionally, laser boriding instead of diffusion boriding was successfully deposited on the pure titanium substrate [18,19]. The direct energy deposition technique was performed to improve the pure titanium properties to be used in different biomedical applications as well. Recently, Bajda et al. [20] deposited S520 bioactive glass coatings onto an ultrafine-grained pure titanium substrate for the first time using the direct energy deposition process, aiming to achieve a toxic-free biomaterial for load-bearing biomedical implants.

More recently, WC-based composites were widely used on Ti-6Al-4V titanium surfaces because of their excellent combination of wear resistance and strength. Al-Sayed et al. [21] studied the deposition of a 60%WC-40%NiCrBSi metal matrix composite layer on a Ti-6Al-4V alloy. They reached an optimal condition at the highest laser heat input (59.5 J·mm^−2^), which achieved the best uniform distribution of WC particles along the deposited layer, with the highest microhardness value of 1200 HV and a notable enhancement in the wear resistance as well. Moreover, Co-based alloy was used as a binding material in some recent studies. Pure titanium (Cp-Ti) and Co powders were deposited at different percentages using an optimized laser processing parameter [22]. Tang et al. [23] added a WC ceramic reinforcement to the Co-based alloy powder, where the hardness increased by 2–4 times over that of the substrate. Furthermore, Wang et al. [24] studied the preparation of pure titanium on Ti-6Al-4V alloy using the direct energy deposition technique. The authors confirmed a significant increase in the hardness and wear resistance after laser processing as compared to the substrate due to three operating strengthening mechanisms: grain refinement, grain boundary, and second-phase strengthening in the clad zone. The residual stresses in the clad zone surface were investigated as well.

Limited studies have considered the conventional treatments of TC21 titanium alloy [25,26,27], whereas scarce studies were reported that investigated laser processing techniques of TC21 alloys, especially direct energy deposition technique. For example, a study on laser peening of TC21 alloy was conducted by Zhigang et al. [28], where this technique induces intensive plastic deformation and high dislocation density to enhance the surface performance of such materials. The results showed an increase in the surface hardness and an improvement in the surface residual stress and fatigue resistance. On the other hand, some studies used the laser additive manufacturing process (LAM) to fabricate a TC21 alloy layer by layer. Chen et al. [29] investigated the microstructure characterizations and the relation between microstructure and thermal history. Another study [30] investigated the tensile properties after the manufacture of different geometries of TC21 under the same processing parameters. The results showed a significant anisotropic tensile property after specific heat treatments. Zhang et al. [31] studied the solidification microstructure and found that it was dominated by columnar grains for a large range of process parameters.

Consequently, from the rather scarce studies reported in this respect, this study was thus planned to deposit a wear-resistant layer to create a uniformly distributed hard surface. Different percentages of WC particles were dispersed in the matrix of the Co-based alloy with the aim of enhancing the poor tribological properties of TC21 titanium alloys. This was achieved by applying the DED process on the substrate surfaces. Constant laser processing parameters were used with different compositions of the deposited powder to reach the optimal condition for better mechanical properties.

## 2. Materials and Methods

### 2.1. Materials

In this study, TC21 titanium alloy in a cylindrical form with a 120 mm diameter and 190 mm length was supplied by Baoji Hanz Material Technology Co., Ltd., Shanxi, China. Its chemical composition is given in Table 1. Small samples with dimensions of 30 × 30 × 10 mm were machined and used as a substrate for the LMD process. All samples were ground with emery paper to remove the oxide scale and rinsed with acetone before starting the DED. Powder of stellite-6 (Co-based alloy) was mixed with spherical fused tungsten carbide particles and used as the deposition powder, as illustrated in Figure 1. Different fractions of WC particles were added to the stellite-6 powder. The particle size of the Co-based alloy ranged from 45 to 150 µm, while the tungsten carbide particle sizes ranged from 40 to 210 µm. The chemical composition of the deposited powder is given in Table 2.

### 2.2. Direct Energy Deposition Setup

The present experiments were conducted on a DED system (TECHNOGENIA Middle East company, Dubai, UAE) that consisted of a 4-kW continuous-wave fiber-coupled diode laser operating at wavelength 1050 nm, a 9-axis numerical control working table, and an attached powder feeding system with a coaxial nozzle. To avoid oxidation, inert gas of pure argon was used during the deposition process. The delivering laser beam had a Gaussian profile with a circular shape of 4 mm diameter. A single-track deposition was performed on the substrate to measure the physical characterization of the deposited layers, while a specific area was covered by applying 8 tracks with 50% overlap to evaluate the tribological properties. A 50% overlap was used to achieve a smooth and homogenous deposited layer; as the laser spot had a 4 mm diameter, we had to move the laser delivery nozzle between 4 mm and 2 mm (0% to 50% overlap). We chose to make a 50% overlap to avoid any discontinuation in the deposited layer [32]. The laser processing parameters of the DED process are listed in Table 3. Three different percentages of WC particles in the stellite-6 powder, as listed in Table 4, were applied in this study to search for an optimal condition for a good metallurgical bond with improved mechanical properties.

### 2.3. Surface Topography and Microstructure Characterization

Metallographic samples were prepared using the standard procedures of mechanical polishing and etching with a solution of 6 mL nitric acid HNO_3_ and 2 mL hydrofluoric acid HF (48% concentration) for 90 s [16]. Microstructures were examined using the Axiotech 30 optical microscope, (Lukas Microscope Service, Inc., Hillview Court, Mundelein, IL, USA), and QUANTA FEG 250 field emission scanning electron microscope (FESEM), Hillsboro, OR, USA, attached with an energy dispersive X-ray (EDX) microanalyzer. The present phases and the new constituents of the deposited layers were analyzed using an XPert Pro analytical X-ray diffractometer (XRD), (D8-Discover-bruker, Karlsruhe, Germany).

### 2.4. Performance Evaluation of the Deposited Layers

#### 2.4.1. Microhardness Test

Vickers microhardness along the cross-section depth of the deposited layers was measured using a Leco LM 7000 micro-hardness tester (Leco Corporation, St. Joseph, MI, USA) under a load of 500 g and a dwell time of 15 s.

#### 2.4.2. Room-Temperature Dry-Sliding Pin-on-Ring Wear Test

A TNO pin-on-ring tribometer (HBM-Hottinger Bladwin Messtechnik GmbH, Darmstadt, Germany) was used to determine the wear resistance of the deposited layers. The wear sample of 7 × 7 × 12 mm in size was fixed against a rotating hardened stainless steel ring with an outer diameter of 73 mm and surface hardness of 63 HRC. The wear test was performed under a load of 40 N and a duration of 30 min, a sliding speed of 0.5 m/s with a total sliding distance of 170 m, and a fixed rotation speed of 150 rpm. The wear weight loss was measured using a high-accuracy photoelectric balance with a 10^−4^ g accuracy. Every experimental test was performed three times to ensure the validity of the obtained results. Worn surfaces were examined after the wear test via optical microscopy.

## 3. Results and Discussion

### 3.1. Characterization of Deposited Layers

#### 3.1.1. As-Cast TC21 Alloy

The as-cast microstructure of the TC21 substrates exhibited a lamellar microstructure, as shown in Figure 2. The structure was composed of a primary alpha phase (α) and the transformed beta (β) matrix [33]. The α phase was distributed homogeneously in the entire field of view. The XRD pattern confirmed the presence of α and β phases in the as-cast TC21 alloy, as shown in Figure 2b.

Commonly, during the DED process, the deposited powder (stellite-6/WC) was melted with a small thickness on the substrate (TC21 alloy) and formed a metal matrix composite (MMC). Various contents of WC particles were found in the deposited layer, according to the different conditions listed previously in Table 4. It is clear from Figure 3 that all deposited layers exhibited a good continuous shape and superior metallurgical bond with the substrate without any signs of surface cracks or lack of adhesion. Consequently, all samples were acceptable for being further investigated.

#### 3.1.2. Sample 1 (100% Stellite-6 Deposited Layer)

The microstructure of sample 1 (S1) consisted of three different zones: deposition zone (DZ), interface zone (IZ), and heat-affected zone (HAZ), as presented in Figure 4. The interface zone (IZ) and heat-affected zone (HAZ) were separated by a fusion line. Higher magnification for each zone is illustrated in Figure 5. The deposition zone showed a hypoeutectic structure of a γ primary Co-rich solid solution, as evidenced from the binary Co–WC phase diagram [34]. Different morphologies of dendritic structures were recorded in the DZ and IZ after the solidification process. This was due to the gradual decrease in the solidification rate and an increase of the temperature gradient through the deposition zone toward the fusion line, which occurred during the DED process. Moreover, the difference in the thermal conductivities led to a thermal gradient between the zones, which resulted in changing the size of the primary solidified dendrite structure in the interface zone, ranging from coarse to fine while approaching the fusion line. The reason was that the thermal gradient depended mainly on the thermal conductivity of the materials and the deposited layer height, which behaved like a heat sink.

In the heat-affected zone (HAZ), the original lamellar (α + β) microstructure of the substrate was solidified as an acicular martensite α’-Ti structure due to the influence of the remelting process.

A compositional EDAX analysis was performed to distinguish the difference in the elemental compositions of the deposited layer, as shown in Figure 6. The deposition zone was observed to contain all elements of the stellite-6 powder plus elemental Ti from the TC21 substrate alloy forming the MMC layer. The distribution of the elements was uniform along the deposition zone. The deposition zone had two different types of clusters: dark gray and white clusters, in addition to the MMC, which was gray. The dark gray phase and white phase contained different percentages of the elements Co, Ti, Cr, C, Ni, and W, as identified in Figure 6d, whereas the white phase consisted mainly of TiC.

The results of the EDAX line scan passing from the deposition zone through the interface zone and reaching the substrate zone are illustrated in Figure 7. Such results confirmed that all elements were evenly distributed along the depth of the MMC layer. As the line scan passed through the matrix, the peaks of the Co and Cr elements increased, whereas the peak of the Ti element was reduced. The opposite occurred when the line scan passed through the interface zone: the Co and Cr elements peaks were lowered, while the Ti peak increased due to the dilution from the substrate. After the observation of these different chemical analyses, an XRD analysis was performed in order to identify all new constituents observed in the deposition layer.

The XRD spectrum of the stellite-6 powder shows the presence of γ-Co, CrO, and Cr_2_C_3_, while the X-ray diffraction results of the deposited layer confirmed the existence of a γ-Co-rich phase, TiC, Co_2_C, Cr_23_C_6_, Co_4_W_2_C, and Ni_3_C carbides, as demonstrated in Figure 8. According to these spectra, it can be concluded that new hard constituents were obtained in the deposited layer. Consequently, by combining the previous EDAX results and the XRD results, the three different phases that were observed in the deposition zone could be identified. The dark grey matrix contained TiC, Co_2_C, Cr_23_C_6_, Co_4_W_2_C, and Ni_3_C; the constituents in the grey matrix were TiC, Co_2_C, and Cr_23_C_6_; and the white areas consisted of TiC, Co_2_C, Cr_23_C_6_, and Co_4_W_2_C.

#### 3.1.3. Sample 2 (60% Stellite-6 Plus 40% Tungsten Carbide (WC) Deposited Layer)

Similarly, the cross-section microstructure of sample 2 could be divided into two main sections separated by the fusion line. The first section contained the deposition zone and the interface zone, whereas the second section consisted of the heat-affected zone and the unaffected base metal zone, as shown in Figure 9.

The MMC microstructure of the deposited layer consisted of a hypoeutectic structure containing a primary γ Co-rich solid solution with a eutectic structure of (γ + WC) particles due to the addition of 40% WC in the deposited powder, as illustrated in Figure 10. The WC particles remained undissolved in the Co-based matrix due to its high melting point (3400 °C). This was in accord with the Guojian et al. study [34]. The point EDAX compositional analysis of the deposited layer confirmed that the deposition zone contained all elements of the stellite-6 powder and the tungsten carbides particles combined with the elements of the TC21 substrate titanium alloy from the substrate (Figure 11), forming the MMC layer. A uniform distribution of the elements was observed along the deposited layer. The presence of the Ti element in the MMC confirmed the good metallurgical bond between the deposited layer and the substrate. In addition, it was expected that thermal stresses and then contraction stresses may have been induced due to the variation of the thermal expansion of each powder [35,36].

The deposition zone shows different particle sizes of undissolved WC particles (bright white or silver; Figure 11a); furthermore, two different cluster types were identified based on the color difference inside the MMC. The MMC was characterized by a gray color that mainly comprised stellite-6 powder elements, whereas the dark gray clusters contained a Co-based alloy with 29% WC and the white color included a high percentage of Ti element in addition to W and C elements, as listed in Figure 11d. The distribution of all elements along the deposited layer is presented in the EDAX line scan shown in Figure 12. Such results confirmed that Co, Cr, Ti, W, and C elements in the deposited layer were uniformly distributed along the MMC depth. Higher W and C percentages were detected due to the addition of 40% WC to the deposited powder. The peaks of the Co, Cr, and W elements increased upon passing over the matrix, whereas the peaks of the Ti element decreased.

Figure 13 presents the X-ray diffraction analysis of sample 2. Additional constituents, such as Co_3_W_3_C and WC, were observed over that detected in sample 1 (recall Figure 8), as 40% of WC was added in the deposited powder. Each constituent in the deposition zone could be identified from the results of both the EDAX and XRD, where the MMC in grey contained TiC, Co, Cr_23_C_6_, Co_3_W_3_C, and WC; the dark grey clusters contained TiC, Co, Cr_23_C_6_, Co_3_W_3_C, and WC; and the white clusters consisted of both TiC and WC. The XRD analysis confirmed that the hypoeutectic structure in the deposition zone consisted of a Co-rich γ phase mixed with complex carbides, such as Cr_23_C_6_, Co_3_W_3_C, and WC.

#### 3.1.4. Sample 3 (40% Stellite-6 Plus 60% Tungsten Carbide (WC) Deposited Layer)

The same zones were observed in the deposited layer of sample 3: deposition zone, interface zone, and the HAZ (see Figure 14). Due to the higher content of WC in the deposited powder, the deposition zone showed a difference in microstructure compared to that obtained from samples 1 and 2. It consisted of a hypereutectic structure: primary tungsten carbide (WC) with a eutectic structure of (γ + WC). The amount of the undissolved WC particles inside the eutectic matrix depended on the weight percent of the WC particles in the deposited powder. The EDAX point analysis confirmed the WC fragmentation inside the deposition zone (Figure 15a), which resulted in the formation of small clusters that mainly consisted of W_2_C; this showed up as the silver clusters illustrated in Figure 16. These clusters resulted from the fragmentation of the WC particles that dispersed homogeneously along the deposition zone. Extra peaks of W_2_C were detected in the XRD data of such samples, in addition to the same peaks of sample 2 (Figure 17). Therefore, the eutectic structure in the deposition zone consisted of a Co-rich γ phase mixed with complex carbides, such as Cr_23_C_6_, Co_3_W_3_C, and W_2_C. Therefore, the hypereutectic structure in the deposition zone consisted of a Co-rich γ phase mixed with complex carbides, such as Cr_23_C_6_, Co_3_W_3_C, and W_2_C.

### 3.2. Microhardness Distribution Profiles

To validate this proposed technique, microhardness measurements were carried out for all deposited layers. Figure 18 displays the microhardness distribution profiles of the three processed samples (1, 2, and 3). The measured hardness reflects the microstructural changes originating from the DED process. As expected, an increase in the hardness occurred from the substrate toward the deposition layer. The microhardness value of the lamellar (α + β) microstructure of the base TC21 alloy was 340 HV. The deposited layer of sample 1 displayed a hardness increase of nearly three times over that recorded for the as-cast sample, the microhardness of the composite matrix in the deposition zone lay between 890 and 1089 HV, while at the interface zone, it ranged from 640 to 680 HV. The hardness of the composite matrix was enhanced by the uniform distribution of the MMC solid solutions and the new forming hard carbides, such as TiC, Co_2_C, Cr_23_C_6_, Co_4_W_2_C, and Ni_3_C. The lower hardness values at the interface zone may be attributed to dilution from the TC21 titanium substrate as was evidenced from the EDAX line analysis in this area (recall Figure 7). Finally, in the heat-affected zone, the hardness reached 470 HV; this emerged from the laser remelting and fast cooling that occurred and resulted in the formation of an α’-Ti martensitic structure.

As concerns samples 2 and 3, a notable increase in microhardness values was observed over that obtained in sample 1. This enhancement was due to the increasing fractions of the hard WC particles in the stellite-6 powder. This resulted in producing a uniform distribution of some new carbides, such as WC, W_2_C, TiC, Cr_23_C_6_, and Co_3_W_3_C, in the MMC solid solution of the deposited layers, as was evidenced by the XRD results.

Referencing sample 2 (Figure 19a), the microhardness of the undissolved WC lay between 1846 and 2410 HV inside the MMC, which showed a hardness range of 880 to 1029 HV. Meanwhile, in the interface zone, the microhardness lay between 650 and 713 HV. On the other hand, for sample 3 (Figure 19b), a further increase in the hardness values was found in the deposition zones due to the increase in the WC content from 40 to 60% of the deposited powder. For the undissolved WC, the hardness lay between 2112 and 2707 HV, while the W_2_C clusters that dispersed in the matrix had hardnesses from 1466 to 2070 HV within the 960 HV MMC. The same hardness values were noticed inside the HAZ (465 HV) for both samples due to the similarity of microstructures. All the above data are summarized in Table 5.

### 3.3. Evaluation of Wear Resistance

The wear properties of the deposited layers with their different WC contents and the commercial TC21 titanium alloy were comparatively investigated. The sliding wear behaviors clearly presented the wear weight loss of the laser-deposited layers as a function of sliding time, as illustrated in Figure 20. The sliding wear depends on numerous factors: (1) substrate properties, such as composition and roughness; (2) deposited powder properties like composition, microstructure, hardness, thickness, and metallurgical bond with TC21 substrate; and (3) wear couple characteristics, such as physical, chemical, and thermal properties, geometrical wear elements, and the type of motion [37,38,39]. Consequently, the deposited layer thicknesses were measured on the deposited single track by means of optical microscopy from the upper point at the external surface to the lower point at the interface zone. The deposited layer measured 1.73 mm for sample 1 and about 1 mm for both samples 2 and 3. As a result, these deposited layers had an adequate thickness to ensure that the measured weight loss after the wear test was from the deposited layer and not from the substrate alloy.

A linear relationship between the weight loss and sliding time was found for the base alloy, reaching a high weight loss after 30 min of sliding time, indicating the rather poor wear resistance for the TC21 alloy. On the other hand, for the laser-processed layers, steady-state wear stages were observed with a slow material removal process and a rather shallow wear rate, particularly for the case of 60% WC. A significant reduction in the wear weight loss amounting to 72%, 93%, and 99% for samples S1, S2, and S3, respectively, after the DED process was recorded relative to that for the as-cast TC21 alloy. This means that increasing the WC percentage in the deposited powder led to further improvement in the wear resistance. The enhancement of the wear resistance of the metal matrix composite layer depends on several factors; these factors may include the uniform distribution of the reinforcements of hard phases, such as WC, W_2_C, TiC, and Cr_23_C_6_, inside the MMC without the existence of any cracks or pores. Additionally, the microhardness value of the deposited layer is another important factor to be considered. In this study, an augmentation in the microhardness value was achieved when compared to the 340 HV of the TC21 alloy, namely, a more than fourfold increase after the DED process. Moreover, the Co-based alloy, which contains chromium, will enhance the resistance to corrosion and high-temperature oxidation, in addition to the strengthening mechanism via the formation of both chromium carbides and titanium carbides, which were successfully obtained after the deposition process [35]. One more factor for the significant enhancement of the wear resistance was the induced compressive residual stresses generated after the very rapid quenching that followed the DED process, as was reported before [40].

In view of the foregoing, the DED process that involved depositing such types of powder on the TC21 titanium alloy under the selected laser processing parameters could be an effective way to extend the service lifetime of TC21 alloys by substantially improving their poor tribological characteristics.

The worn surfaces of the starting material, as well as the DED samples, were examined by means of optical microscopy, as shown in Figure 21. Wear debris exposed the effects of several features, such as the test load, the ring type, and the type of the deposited powder in the DED process. The worn surface of the as-cast TC21 exhibited plough grooves, while the material removal mechanism was adhesive for the MMC surface of samples with 100% stellite-6 powder due to the abrasion of hardened SiC ring on the soft titanium and MMC surface. In contrast, for samples S2 (60% stellite-6 and 40% WC) and S3 (40% stellite-6 and 60% WC), the material removal mechanism changed from ploughing to an adhesion wear mechanism with a decrease in the ploughing groove with an increase in the percentage of WC in the deposited powder, as shown in the surface morphologies of the worn surfaces in Figure 21c,d. Consequently, the deposited layer considerably protected the TC21 alloy from severe wear and created a smooth surface with very fine wear debris. Although the wear resistance of the deposited material is remarkably enhanced, the location of WC particles on the worm surface has to be further investigated as this behavior is expected to affect the fatigue performance for this material.

## 4. Conclusions

Depositions of the mixture powder of Co-based alloy (stellite-6) and up to 60 wt.% of WC on TC21 titanium alloy plates were successfully performed using a direct energy deposition process with a constant laser processing parameter.

The so-obtained results may be summarized in the following:The microstructure of the deposited layer consists of a hypoeutectic or hypereutectic structure with undissolved tungsten carbide particles dispersed in the Co-based alloy matrix that depended on the content of WC.Additional carbides, such as TiC, Cr_23_C_6_, Co_2_C, Co_4_W_2_C, Co_3_W_3_C, WC, and W_2_C, were also observed in the deposition zone.The microhardness level continuously rose through the substrate alloy toward the deposition layer due to the change of microstructure. The hardness value of sample 1 with no WC addition was enlarged by nearly three times over the substrate hardness, whereas a notable increase (fourfold) in the microhardness values for sample 2 (60% stellite-6 plus 40% tungsten carbide (WC) deposited layer) and sample 3 (60% stellite-6 plus 40% tungsten carbide (WC) deposited layer)) were recorded due to the addition of different percentages of the hard WC particles.The wear resistance of the deposited layer was significantly improved, as reflected by a decrease in the samples’ wear weight loss by 72%, 93%, and 99% for samples S1, S2, and S3, respectively, over that obtained from the substrate alloy.The worn surface of the TC21 alloy presented plough grooves, while the material removal mechanism was adhesive for the MMC surface of samples with 100% stellite-6 powder; for samples S2 (60% stellite-6 and 40% WC) and S3 (40% stellite-6 and 60% WC), the material removal mechanism changed from ploughing to an adhesion wear mechanism.

## Figures and Tables

**Figure 1 micromachines-12-00739-f001:**
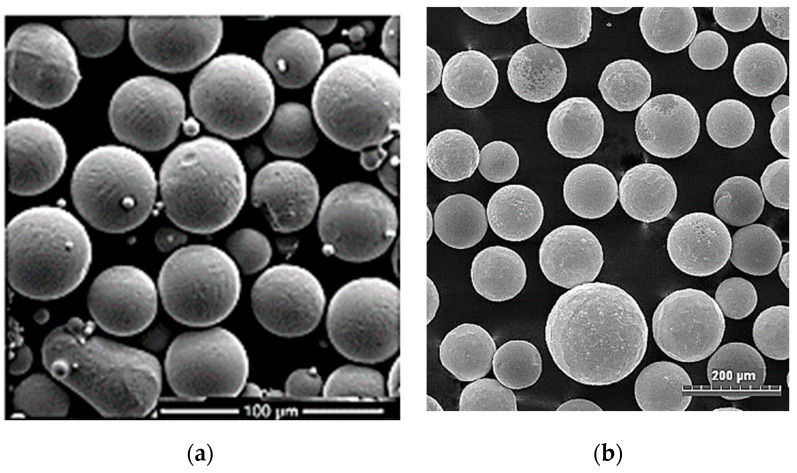
FESEM powder micrographs of (**a**) stellite-6 and (**b**) spherical tungsten carbides.

**Figure 2 micromachines-12-00739-f002:**
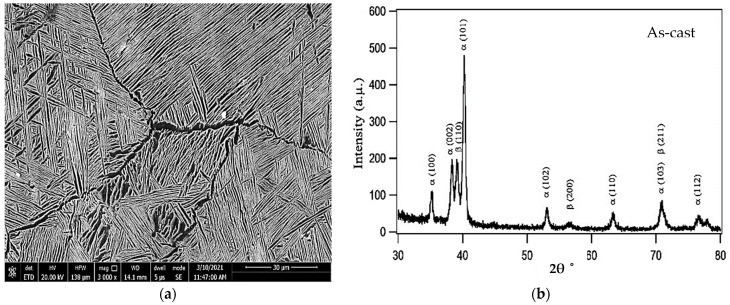
(**a**) FESEM micrograph of the as-cast TC21 alloy and (**b**) XRD pattern of the as-cast TC21 alloy.

**Figure 3 micromachines-12-00739-f003:**
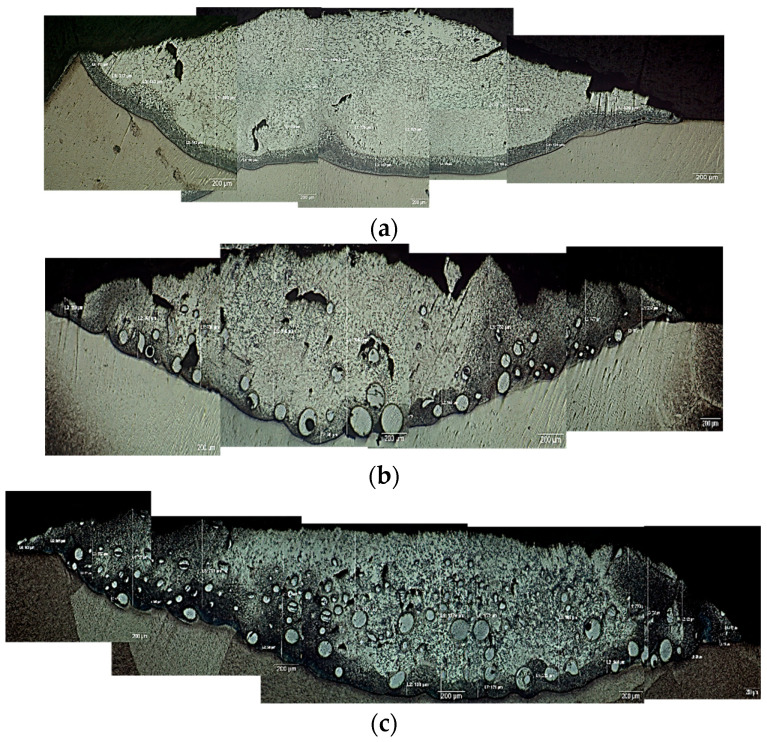
Light micrographs of cross-sections of the deposited layers of (**a**) sample 1, (**b**) sample 2, and (**c**) sample 3.

**Figure 4 micromachines-12-00739-f004:**
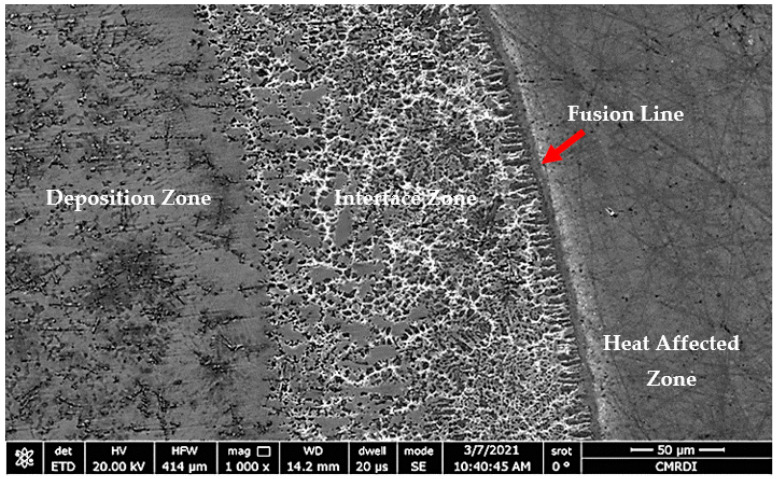
FESEM micrograph of S1 cross-section of the whole deposited layer showing three different zones: deposition, interface, and HAZ.

**Figure 5 micromachines-12-00739-f005:**
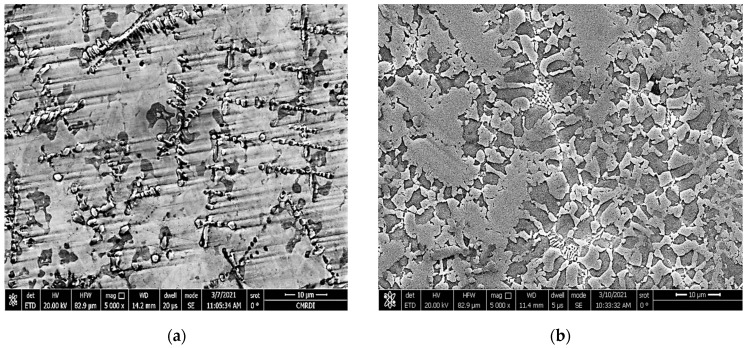
FESEM micrographs of the (**a**) hypoeutectic structure in the DZ and (**b**) dendritic structure of the IZ.

**Figure 6 micromachines-12-00739-f006:**
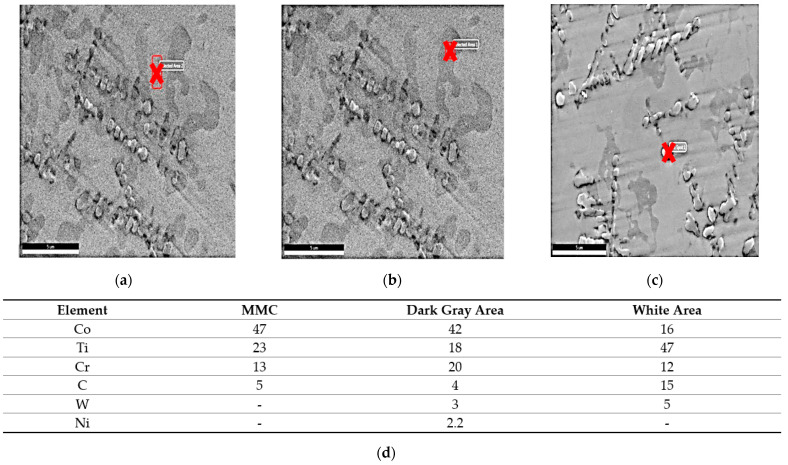
SEM micrographs and EDAX point analysis results of the deposited layer made with 100% stellite-6 powder: (**a**) gray, (**b**) dark gray, (**c**) white, and (**d**) the chemical analysis results for each color.

**Figure 7 micromachines-12-00739-f007:**
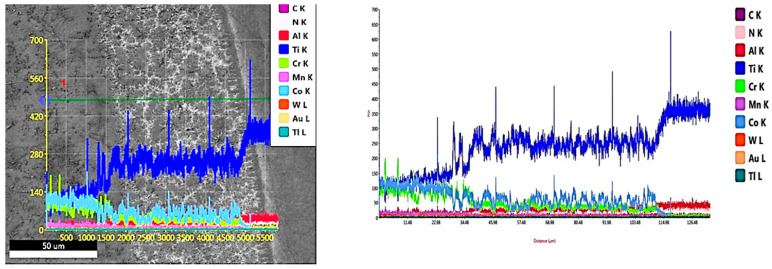
EDAX line scan along the cross-section of MMC layer of sample 1.

**Figure 8 micromachines-12-00739-f008:**
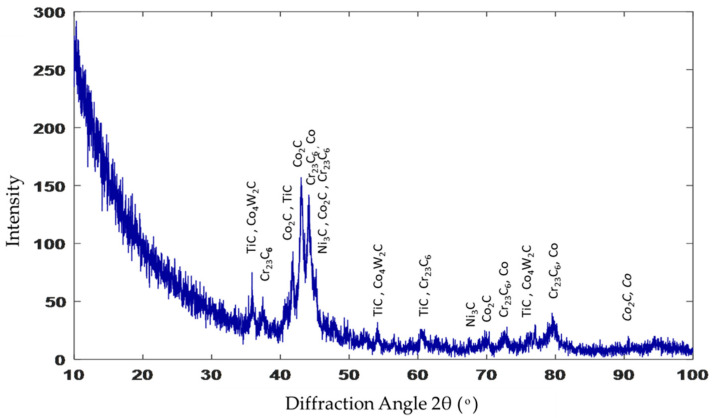
XRD pattern of sample 1.

**Figure 9 micromachines-12-00739-f009:**
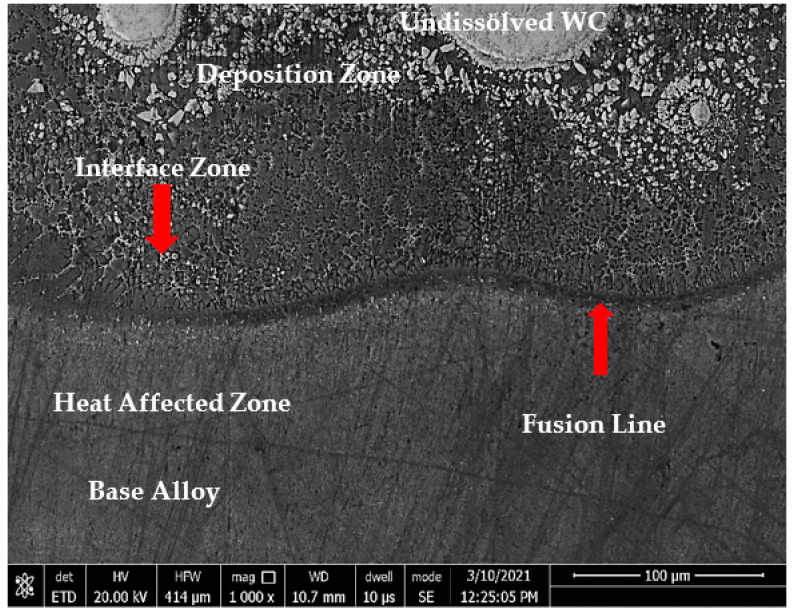
FESEM micrograph of a cross-section of S2 across the whole deposited layer showing three different zones: deposition, interface, and HAZ.

**Figure 10 micromachines-12-00739-f010:**
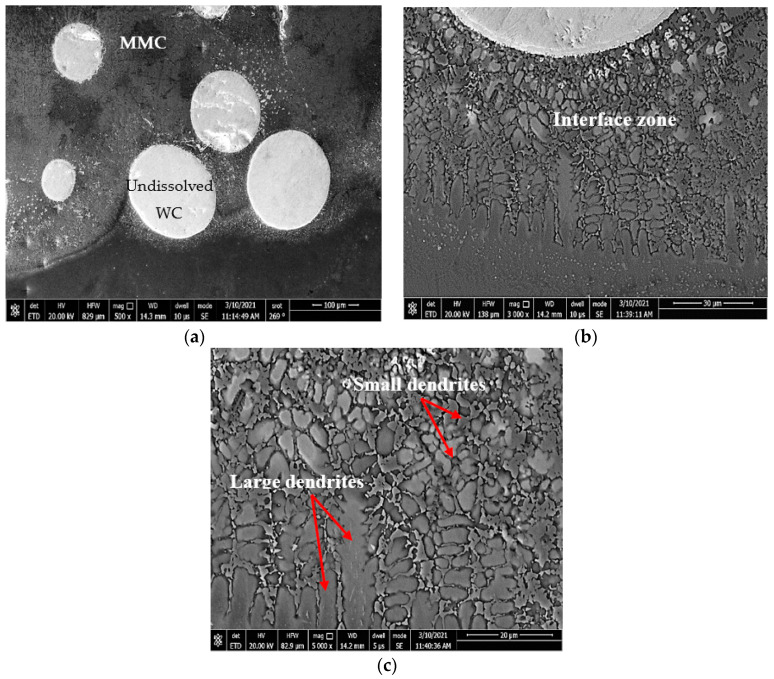
FESEM micrographs of the (**a**) hypoeutectic structure at the DZ, (**b**) dendritic structure at the IZ, and (**c**) large dendrite structure close to the fusion line.

**Figure 11 micromachines-12-00739-f011:**
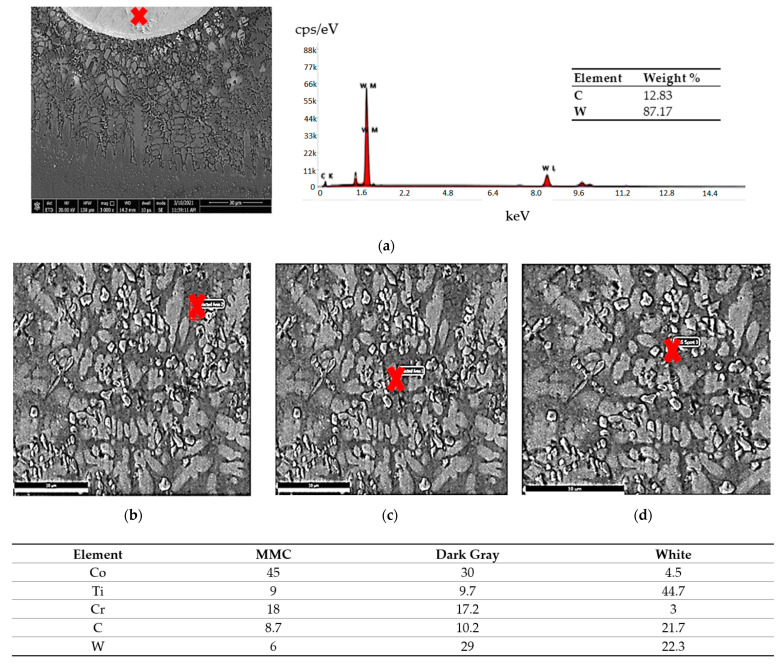
EDAX point analysis results of the deposited layer made with 60% stellite-6 powder + 40% WC: (**a**) WC particles, (**b**) (MMC) gray, (**c**) dark gray, (**d**) white, and (**e**) the chemical analysis of each constituent.

**Figure 12 micromachines-12-00739-f012:**
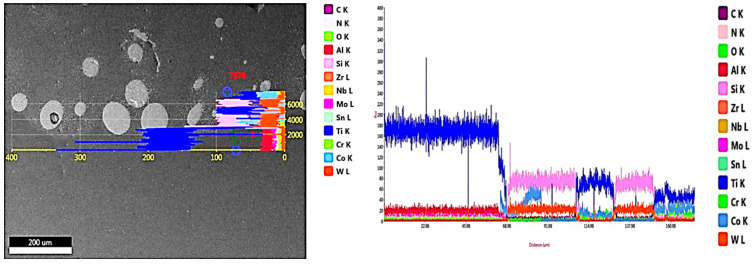
EDAX line scan along the cross-section of the MMC layer of sample 2.

**Figure 13 micromachines-12-00739-f013:**
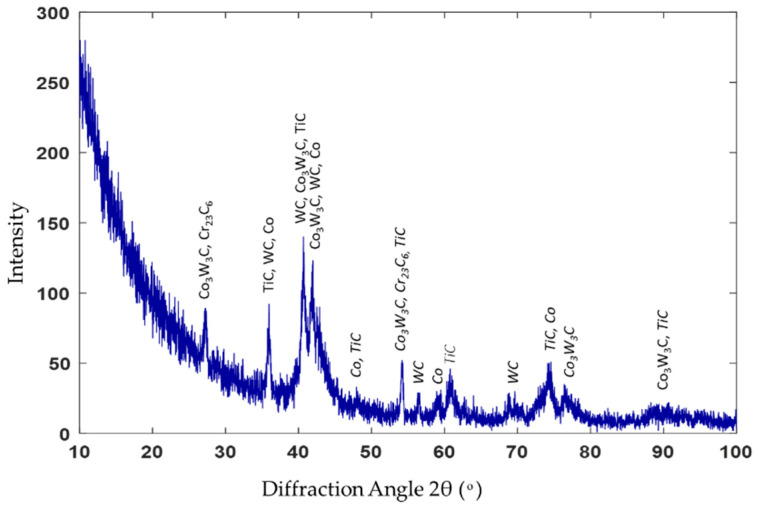
XRD spectrum of sample 2.

**Figure 14 micromachines-12-00739-f014:**
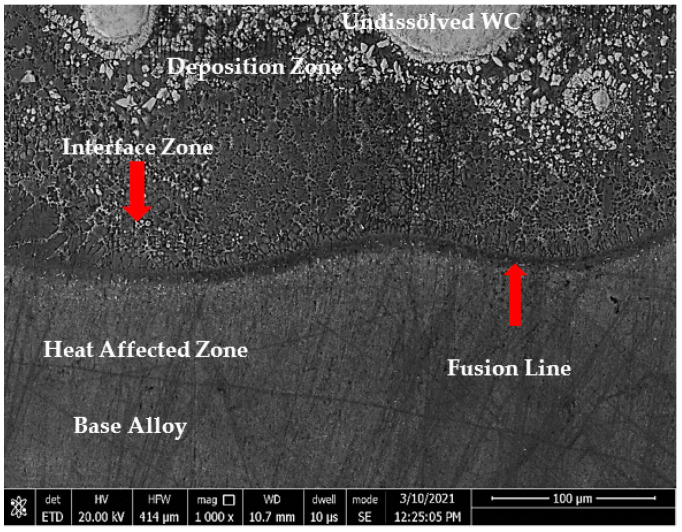
FESEM micrograph of a cross-section of sample 3’s whole deposited layer showing three different zones: deposition, interface, and HAZ.

**Figure 15 micromachines-12-00739-f015:**
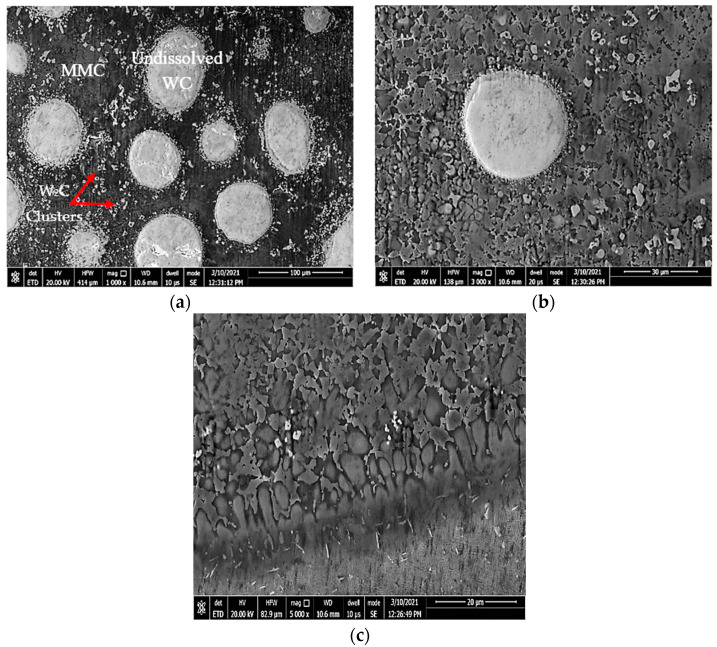
FESEM micrographs of the (**a**) hypereutectic structure at the DZ, (**b**) dendritic structure at the IZ, and (**c**) large dendrite structure close to the fusion line.

**Figure 16 micromachines-12-00739-f016:**
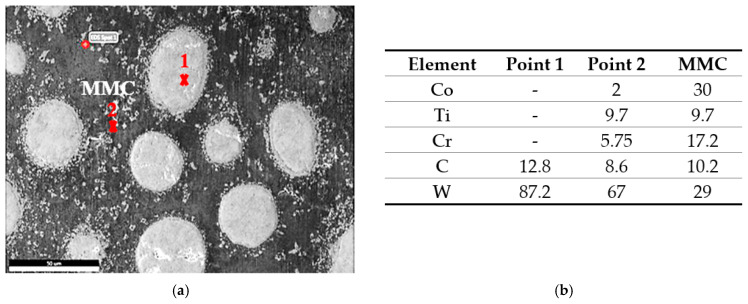
EDAX point analysis results of the deposited layer made with 40% stellite-6 powder + 60% WC: (**a**) SEM micrograph of deposited layer, and (**b**) the chemical analysis of each constituent.

**Figure 17 micromachines-12-00739-f017:**
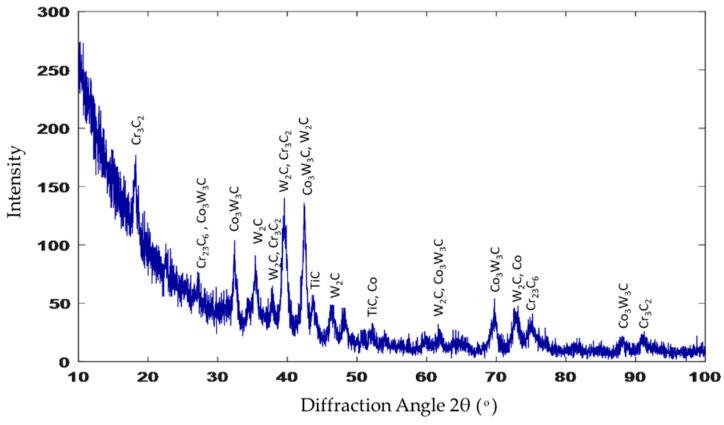
XRD spectrum of sample 3.

**Figure 18 micromachines-12-00739-f018:**
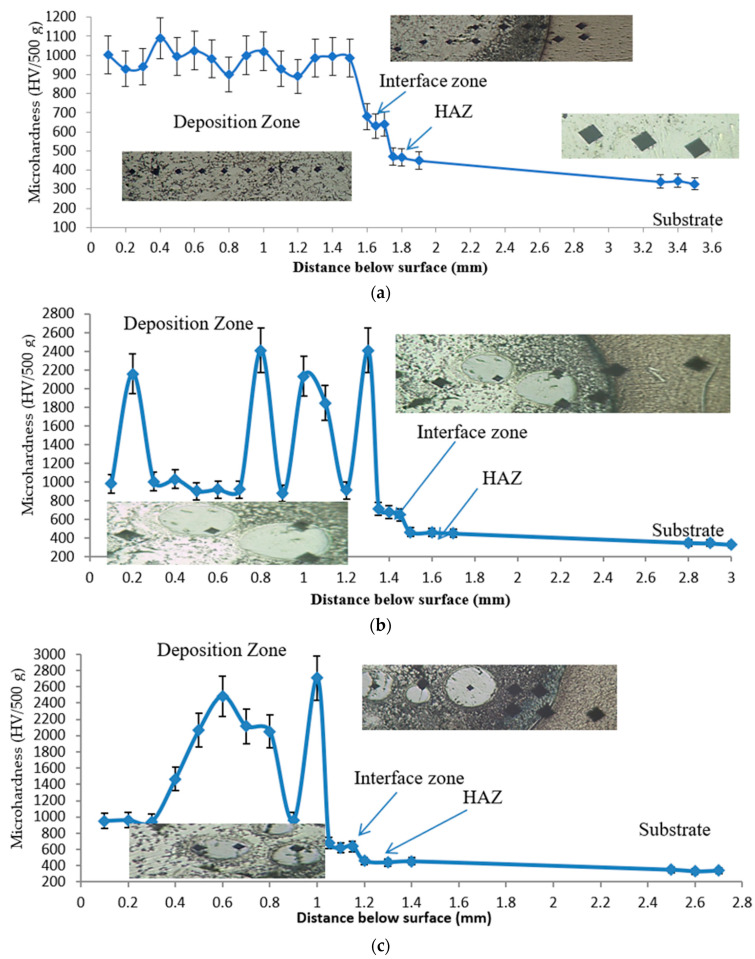
Microhardness profiles along the different zones (DZ, IZ, and HAZ) of the deposited layer of (**a**) sample 1, (**b**) sample 2, and (**c**) sample 3.

**Figure 19 micromachines-12-00739-f019:**
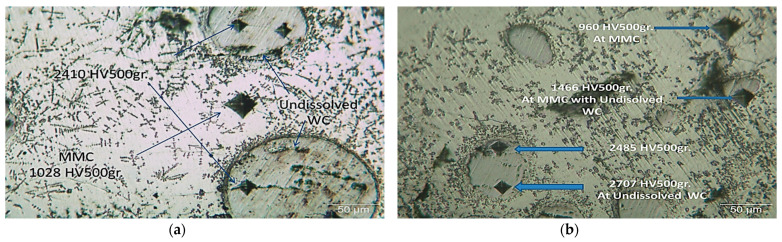
Microhardness variations in the deposition zone for (**a**) sample 2 and (**b**) sample 3.

**Figure 20 micromachines-12-00739-f020:**
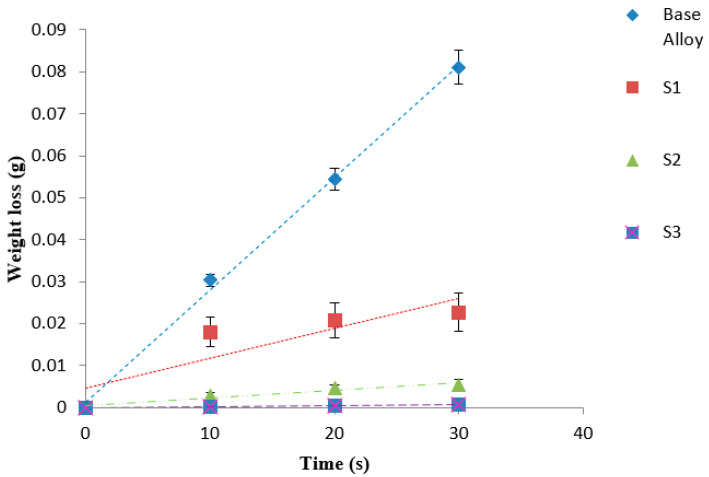
Weight loss of as-cast alloy and the deposited layer of samples 1, 2, and 3.

**Figure 21 micromachines-12-00739-f021:**
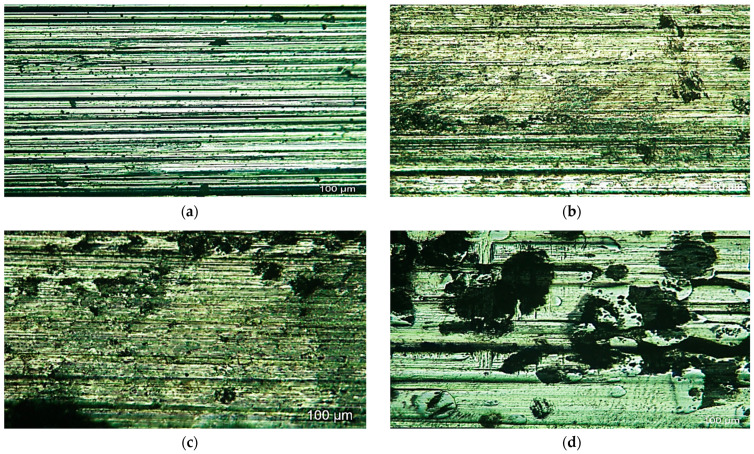
Optical images of worn surface morphologies of (**a**) as-cast sample, (**b**) sample 1, (**c**) sample 2, and (**d**) sample 3.

**Table 1 micromachines-12-00739-t001:** Chemical composition of TC21 titanium alloy (wt.%).

Element	Al	Mo	Sn	Zr	Nb	Cr	Si	Ti
Concentration	6.5	3.0	2.2	2.2	1.9	1.5	0.09	Balance

**Table 2 micromachines-12-00739-t002:** Chemical composition of the deposited powder (wt.%).

Powder Type	Concentration (wt.%)								
Stellite-6	C	Mo	W	Ni	Mn	Cr	Si	Fe	Co
	1.4	1.0	4.8	3.0	1.0	29.5	1.3	3.0	Balance
WC	C		Fe		Ti		W		
	4.0		0.23		0.05		Balance		

**Table 3 micromachines-12-00739-t003:** Experimental parameters of the DED process.

Parameters	Value
Laser power	2000 W
Scanning speed	900 mm/min
Deposition rate	20 g/min
Feeding gas	5 L/min
Shielding gas	15 L/min
Defocus distance	16 mm
Beam diameter	4 mm

**Table 4 micromachines-12-00739-t004:** Data of deposited powder on TC21 substrates.

Powder Composition (%)	Sample 1	Sample 2	Sample 3
Stellite-6	100	60	40
Tungsten carbide	0	40	60

**Table 5 micromachines-12-00739-t005:** Summarized microhardness data of samples 1, 2, and 3.

Zone Name	Sample 1	Sample 2	Sample 3
MMC (DZ + WC particles)	890 to 1089 HV	880 to 1029 HV with WC particles of 1846 to 2410 HV	960 HV with WC particles of 2112 to 2707 HV and W_2_C clusters of 1466 to 2070 HV
IZ	640 to 680 HV	650 to 713 HV	750 to 810 HV
HAZ	470 HV		
Substrate alloy	340 HV		

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
