# Peer review of "Laser Surface Modification of TC21 (α/β) Titanium Alloy Using a Direct Energy Deposition (DED) Process"

_micromachines, 2021, doi:10.3390/mi12070739_

Round 1
Reviewer 1 Report
The article uses 'laser surface modification', 'laser cladding' and ' Laser Metal Deposition'. Please stick to just one term throughout the article. Suggest to use Diected Energy Deposition (DED), as this is an ASTM standard.
Line 44: use weight reduction instead.
Line 70: toxic-free
Replace Fig 1 (a) to a high quality micrograph image at the same scale as image in (b).
Line 135: explain why 50% overlap was used.
Author Response
Response to Reviewer Comments
Dear Reviewer (1)
We would like to thank you for giving us a constructive suggestions which would help us to improve the quality of the article (Manuscript ID: micromachines-1259289). Here we submit a revised version of our manuscript entitled “Laser Surface Modification of TC21 (α/β) Titanium Alloy Using Direct Energy Deposition (DED) Process”, which has been modified according to your suggestions.
The following is a point-to-point response to your comments. All revisions are be clearly highlighted using the "Track Changes" function in Microsoft Word, so that changes are easily visible to the editor and all reviewers.
Detailed Letter of Response of Reviewer # 1
The article uses 'laser surface modification', 'laser cladding' and ' Laser Metal Deposition'. Please stick to just one term throughout the article. Suggest to use Directed Energy Deposition (DED), as this is an ASTM standard.
- Responses according to the reviewer’s comments:
- Line 44: use weight reduction instead.
- Thanks for your valuable comment.
- This sentence has been modified as follows:
“The use of TC21 alloy in aircraft components such as landing gear or flap track, there will be a significant advantage on the weight reduction for such aircraft [9].”
- Line 70: toxic-free
- Thanks for your important comment.
- This sentence has been modified as follows:
“Recently, Bajda et al., [20] deposited S520 bioactive glass coatings onto an ul-trafine-grained pure titanium substrate for the first-time using direct energy deposition process, aiming to achieve toxic-free biomaterial for load bearing biomedical im-plants.”
- Replace Fig 1 (a) to a high quality micrograph image at the same scale as image in (b).
- Thanks for your constructive comment.
- It has been modified and most of figures have been improved.
- Line 135: explain why 50% overlap was used.
- Thanks for your constructive comment.
- This part has been added as follows:
“A 50% overlap was used to achieve a smooth and homogenous deposited layer, as the laser spot has 4 mm diameter, we had to move the laser delivery nozzle between 4 mm and 2 mm (0% to 50% overlap). We chose to make a 50% overlap to avoid any discon-tinuation in the deposited layer [32].”
Hoping that the changes introduced improved the manuscript in satisfactory way. With our best regards

Reviewer 2 Report
The paper “Laser surface modification of TC21 (alpha/beta) Titanium alloys using laser metal deposition (LMD) Process” evaluated the effect of depositing WC particles dispersed in stellite-6 on TC21 substrates concerning mechanical properties (microhardness and wear resistance), and microstructural features (analysed via SEM, EDX and XRD). The quality of the work is high. Good explanations on methods and results can be found. Some small comments must be addressed, but in general the paper is very good. Congratulations.
- On line 37, the authors describe the main characteristics of titanium-alloys which promotes their use in aerospace applications, aero-engines, and chemical industries. One of the features highlighted is “excellent biocompatibility”. Could the authors discuss why the aforementioned industries are interested in biocompatibility?
- Check English consistency on line 40.
- What is ‘Cp-Ti’ on line 78?
- Check English consistency on lines 94-98.
- Do the authors mean “samples” instead of “surfaces” on lines 113 and 115?
- Please improve the quality of Fig. 5
- Lines 259-260 the authors comment that thermal stresses (contraction) may be induced due to thermal conductivity mismatch among the powders. Wouldn’t it be related to thermal expansion mismatch instead?
- Verify subscript on line 303.
- What ‘()’ means on line 374?
- Typo on line 399. Figure 21 must be replaced by Figure 11.
- Typo on line 407. Figure 22 must be replaced by Figure 11.
- Please re-write the first paragraph of the conclusions and better describe the type of coatings analysed in the study.
- In the conclusions, please describe the composition of samples 1, 2 and 3. It is known that there is an explanation throughout the text, but it is a good practice to remind the reader in the conclusions.
- When the authors describe the wear behaviour of the distinct samples, it is clear that although the WC+stellite-6 coated samples present reduced wear, the wear location is apparently concentrated in the WC previous location. How can this influence the fatigue behaviour of the coated material, since superficial defects are being induced in the sample. This does not seem to be the case for the 'as-received' sample and sample 1.
- The quality of the Figures could be improved.
Author Response
Response to Reviewer Comments
Dear Reviewer (2)
We would like to thank you for giving us a constructive suggestions which would help us to improve the quality of the article (Manuscript ID: micromachines-1259289). Here we submit a revised version of our manuscript entitled “Laser Surface Modification of TC21 (α/β) Titanium Alloy Using Direct Energy Deposition (DED) Process”, which has been modified according to your suggestions.
The following is a point-to-point response to your comments. All revisions are be clearly highlighted using the "Track Changes" function in Microsoft Word, so that changes are easily visible to the editor and all reviewers.
Detailed Letter of Response of Reviewer # 2
The paper “Laser surface modification of TC21 (alpha/beta) Titanium alloys using laser metal deposition (LMD) Process” evaluated the effect of depositing WC particles dispersed in stellite-6 on TC21 substrates concerning mechanical properties (microhardness and wear resistance), and microstructural features (analysed via SEM, EDX and XRD). The quality of the work is high. Good explanations on methods and results can be found. Some small comments must be addressed, but in general the paper is very good. Congratulations.
- Responses according to the reviewer’s comments:
- On line 37, the authors describe the main characteristics of titanium-alloys which promotes their use in aerospace applications, aero-engines, and chemical industries. One of the features highlighted is “excellent biocompatibility”. Could the authors discuss why the aforementioned industries are interested in biocompatibility?
Response:
- Thanks for your constructive comment.
- This part of “excellent biocompatibility” has been removed because we focused on improving properties which is important for aerospace applications, aero-engines, and chemical industries.
“They are widely used in advanced aerospace applications, aero-engines, and chemical industries, due to their unique merits of low density, the combination of high strength-to-weight ratio heat resistance, and corrosion resistance [4,5].”
- Check English consistency on line 40.
- Thanks for your constructive comment.
- This part has been modified and whole manuscript has been carefully checked by a native English-speaking colleague.
“Recently, the TC21 titanium alloy (Ti-6Al-2Sn-2Zr-3Mo-1Cr-2Nb-Si, wt.%) is considered as a new version in aerospace applications that replaced the commercial Ti6Al4V grade five titanium alloys. It belongs to the family of (α+β) titanium alloys with a high strength, toughness, and damage-tolerance properties [8].”
- What is ‘Cp-Ti’ on line 78?
- Thanks for your constructive comment.
- This part has been modified to identify ‘Cp-Ti’ as follows:
“Moreover, Co-based alloy was used as a binding material in some recent studies. Pure ti-tanium (Cp-Ti) and Co powders were deposited at different percentages by an optimized laser processing parameter [22].”
- Check English consistency on lines 94-98.
- Thanks for your constructive comment.
- This part has been modified and whole manuscript has been carefully checked by a native English-speaking colleague.
“On the other hand, some studies used laser additive manufacturing process (LAM) to fab-ricate the TC21 alloy layer by layer. Chen et al., [29] investigated the microstructure char-acterizations and the relation between microstructure and thermal history. Another re-search [30] studied the tensile properties after the manufacture of different geometries of TC21 under the same processing parameters. The results showed significant anisotropic tensile property after specific heat treatments. Zhang et al., [31] studied the solidification microstructure and found that it was dominated by columnar grains for a large range of process parameters.”
- Do the authors mean “samples” instead of “surfaces” on lines 113 and 115?
- Thanks for your constructive comment.
- This part has been modified as follows:
“Small samples with dimension of 30 mm × 30 mm × 10 mm were used as a substrate for the LMD process. All samples were ground with emery paper to remove the oxide scale and rinsed with acetone before starting the LMD.”
- Please improve the quality of Fig. 5
- Thanks for your constructive comment.
The figure quality has been improved as follows:
Figure 1. FESEM micrographs of the (a) hypoeutectic structure in the DZ and (b) dendritic structure of the IZ.
- Lines 259-260 the authors comment that thermal stresses (contraction) may be induced due to thermal conductivity mismatch among the powders. Wouldn’t it be related to thermal expansion mismatch instead?
- Thanks for your constructive comment.
- This part has been modified as follows:
“In addition to, it is expected that thermal stresses and then contraction stresses may be in-duced due to the variation of the thermal expansion of each powder [35,36].”
- Verify subscript on line 303.
- Thanks for your valuable comment.
- This part has been subscripted.
- What ‘()’ means on line 374?
- This “()” has been removed.
- Typo on line 399. Figure 21 must be replaced by Figure 11.
- Thanks for your valuable comment.
- It has been modified and all serial numbers of all Figures have been checked.
- Typo on line 407. Figure 22 must be replaced by Figure 11.
- Thanks for your valuable comment.
- It has been modified and all serial numbers of all Figures have been checked.
- Please re-write the first paragraph of the conclusions and better describe the type of coatings analysed in the study.
- Thanks for your valuable comment.
- This part has been re-written as follows:
“A deposition of the mixture powder of Co-based alloy (Stellite-6) and up to 60 wt.% of WC on TC21 titanium alloy plates was successfully performed using a direct energy dep-osition process with constant laser processing parameter.
The so-obtained results may be summarized in the following:”
- In the conclusions, please describe the composition of samples 1, 2 and 3. It is known that there is an explanation throughout the text, but it is a good practice to remind the reader in the conclusions.
- Thanks for your valuable comment.
- This part has been modified as follows:
“The microhardness level continuously rises through the substrate alloy towards the depo-sition layer due to the change of microstructure. The hardness value of Sample 1, with no WC addition was enlarged by nearly 3 times over the substrate hardness, whereas a nota-ble increase (fourfold) in the microhardness values for sample 2 (60% stellite-6 plus 40% tungsten carbide (WC) deposited layer) and sample 3 (60% stellite-6 plus 40% tungsten carbide (WC) deposited layer)) were recorded due to addition of different percentages of the hard WC particles.”
- When the authors describe the wear behaviour of the distinct samples, it is clear that although the WC+stellite-6 coated samples present reduced wear, the wear location is apparently concentrated in the WC previous location. How can this influence the fatigue behaviour of the coated material, since superficial defects are being induced in the sample. This does not seem to be the case for the 'as-received' sample and sample 1.
- Thanks for your valuable comment.
- This part has been added in section 3.3 as follows:
“Although, the wear resistance of the deposited material is remarkably enhanced, yet the location of WC particles on the worm surface has to be further investigated as this behavior is expected to affect the fatigue performance for this material.”
- The quality of the Figures could be improved.
- Thanks for your valuable comment.
- Most of figures have been improved.
Hoping that the changes introduced improved the manuscript in satisfactory way. With our best regards

Reviewer 3 Report
Review Comments on the Manuscript micromachines-1259289
This paper is very interesting. Laser techniques are used in many industries. Laser technology enables precise machining or micro-machining of material and surfaces thanks to the use of a laser beam as a non-contact, automated tool for shaping them. This study aimed to improve the mechanical properties of chosen alloys. Laser surface modification is applied to enhance the hardness, tribological properties and corrosion resistance for TC21 (α/β) Titanium alloys.
Editing
- Page 1, line 13- there is “tribological properties and”, but should be “, tribological properties, and”.
- Page 1, line 16 - there is “ TC21 by means of 4 KW continuous wave fiber-coupled diode laser at constant”, but should be “TC21 using a 4 KW continuous-wave fiber-coupled diode laser at a constant ”.
- Page 1, line 18 - there is “ This study aims to obtain a uniform distribution of hard surface”, but should be” This study aims to obtain a uniform distribution of hard surfaces”.
- Page 1, line 18 - there is “WC particle”, but should be” WC particles”.
- Page 1, line 20 - there is “ (EDAX) and X-ray”, but should be”(EDAX), and X-ray”.
- Page 1, line 23 - there is “temperature by means of”, but should be” temperature using dray”.
- Page 1, line 24 - there is “microstructure of deposited layer consisted of hypereutectic structure”, but should be” microstructure of the deposited layer consisted of a hyper eutectic structure”.
- Page 1, line 27 - there is “with the as received samples.”, but should be ”as-received samples”.
- Page 1, line 32 - there is “titanium and their alloys”, but should be ”titanium and its alloys”.
- Page 1, lines 36,37 - there is “and excellent biocompatibility”, but should be ” and excellent biocompatibility”.
- Page 1, line 37 - there is “titanium makes”, but should be ” titanium make”.
- Page 1, line 40 - there is “which replaced”, but should be ” that replaced”.
- Page 1, line 42 - there is “In case of”, but should be ”In the case of”.
- Page 1, line 44 - there is “aircrafts”, but should be ” aircraft”.
- Page 2, line 51 - there is “[11].Laser”, but should be ” [11]. Laser”.
- Page 3, lines 111, 112 - the number and the unit should be written in one line.
- Page 4, line 141 - there is “Table 4. conditions of deposited powder on TC21 substrates.”, but should be ” Table 4. Conditions of deposited powder on TC21 substrates.”.
- The numbering of the figures is incorrect in the text. For example, figure 2 is on the page 5 and 7.
- Lines 225, 278 - poorly visible markers in figures a, b, c. This should be corrected.
- Lines 237,291,315 - the font size in the figure should be unicolative.
- Line 278 - in figure a, the x and y axes should be described.
- Line 311- the figure should be read in its entirety on the second page. This should be corrected.
- Line 339 – 340 - the number should be written together with the unit on one line.
- Line 374- there is” ()”, ????.
- Lines 390-391 - it should be written on one line “process 390 [33]”.
- Line 410 - Delete the border from the figure.
- Line 412 - In Figure 11b and 11d a faint marker. This should be reinforced.
- Line 457 - a year is missing.
Substantive
- The article lists some minor linguistic errors on the first page of the article, but it is recommended to correct it throughout the text.
- When there is a reference to a figure in a sub-point in the text, the drawing should be in the same paragraph and not elsewhere (for example: line 180 – Fig. 3. Page 5, line 203, page 6 - another paragraph).
- The numbering of figures and their references should be confirmed throughout the text. For example, there is no figure 7.
- Apart from the SEM and XRD studies, there is no in-depth analysis using TEM.
- No statistics (e.g. standard deviation) compiled for the measurements taken on the tested samples. The statistics for the following investigations: dry sliding pin-on-ring wear test, microhardness test should be complied in paper.
Therefore, I recommend that this manuscript consider publication, after taking into account the editing and substantive corrections.
Author Response
Response to Reviewer Comments
Dear Reviewer (3)
We would like to thank you for giving us a constructive suggestions which would help us to improve the quality of the article (Manuscript ID: micromachines-1259289). Here we submit a revised version of our manuscript entitled “Laser Surface Modification of TC21 (α/β) Titanium Alloy Using Direct Energy Deposition (DED) Process”, which has been modified according to your suggestions.
The following is a point-to-point response to your comments. All revisions are be clearly highlighted using the "Track Changes" function in Microsoft Word, so that changes are easily visible to the editor and all reviewers.
Detailed Letter of Response of Reviewer # 3
This paper is very interesting. Laser techniques are used in many industries. Laser technology enables precise machining or micro-machining of material and surfaces thanks to the use of a laser beam as a non-contact, automated tool for shaping them. This study aimed to improve the mechanical properties of chosen alloys. Laser surface modification is applied to enhance the hardness, tribological properties and corrosion resistance for TC21 (α/β) Titanium alloys.
- Responses according to the reviewer’s comments:
Editing:
- Page 1, line 13- there is “tribological properties and”, but should be “, tribological properties, and”.
- Thanks for your constructive comment.
- This sentence has been modified as follows:
“Recently, direct energy deposition is usually applied to enhance the hardness, tribological properties, and corrosion resistance for many alloys.”
- Page 1, line 16 - there is “ TC21 by means of 4 KW continuous wave fiber-coupled diode laser at constant”, but should be “TC21 using a 4 KW continuous-wave fiber-coupled diode laser at a constant ”.
- Thanks for your constructive comment.
- This sentence has been modified as follows:
“Different WC percentages were applied on the surfaces of TC21 using a 4 KW continuous-wave fiber-coupled diode laser at a constant powder feeding rate.”
- Page 1, line 18 - there is “ This study aims to obtain a uniform distribution of hard surface”, but should be” This study aims to obtain a uniform distribution of hard surfaces”.
- Thanks for your constructive comment.
- This sentence has been modified as follows:
“This study aims to obtain a uniform distribution of hard surfaces”
- Page 1, line 18 - there is “WC particle”, but should be” WC particles”.
- Thanks for your constructive comment.
- This sentence has been modified as follows:
“containing undissolved WC particles dispersed in a Co-based alloy matrix to enhance the wear resistance of such alloys.”
- Page 1, line 20 - there is “ (EDAX) and X-ray”, but should be”(EDAX), and X-ray”.
- Thanks for your constructive comment.
- This sentence has been modified as follows:
“Scanning electron microscopy, energy dispersive X-ray analyzer (EDAX), and X-ray diffractometry (XRD) were used to characterize the deposited layers. New constituents and intermetallic compounds were found in the deposited layers.”
- Page 1, line 23 - there is “temperature by means of”, but should be” temperature using dray”.
- Thanks for your constructive comment.
- This sentence has been modified as follows:
“Microhardness was measured for all deposited layers and wear resistance was evaluated at room temperature using dry sliding ball on disk abrasion test.”
- Page 1, line 24 - there is “microstructure of deposited layer consisted of hypereutectic structure”, but should be” microstructure of the deposited layer consisted of a hyper eutectic structure”.
- Thanks for your constructive comment.
- This sentence has been modified as follows:
“The results showed that the microstructure of the deposited layer consisted of a hypereutectic structure and undissolved tungsten carbide dispersed in the matrix of the Co-based alloy depending on the percentage of WC weight fraction.”
- Page 1, line 27 - there is “with the as received samples.”, but should be ”as-received samples”.
- Thanks for your constructive comment.
- Because it is a cast material not as-received one, as-cast has been witten instead of as-received throughout the whole manuscript.
- This sentence has been modified as follows:
“The microhardness values increased with raising the WC weight fraction in the deposited powder by more than three folds as compared with the as-cast samples. A notable enhancement of wear resistance of the deposited layers was thus achieved.”
- Page 1, line 32 - there is “titanium and their alloys”, but should be ”titanium and its alloys”.
- Thanks for your constructive comment.
- This sentence has been modified as follows:
“Nowadays titanium and its alloys particularly have received more attention for a wide range of applications in many fields such as military, medical, and civil transporta-tion, especially α+β titanium alloys [1–3].”
- Page 1, lines 36,37 - there is “and excellent biocompatibility”, but should be ” and excellent biocompatibility”.
- Thanks for your constructive comment.
- This sentence has been removed as a response to reviewer # (2)
- Page 1, line 37 - there is “titanium makes”, but should be ” titanium make”.
- Thanks for your constructive comment.
- This sentence has been modified as follows:
“Therefore, all these characteristics make them the most favorable material choices for cer-tain applications such as airplanes making up to 30% -50% weight of the total structure [6,7].”
- Page 1, line 40 - there is “which replaced”, but should be ” that replaced”.
- Thanks for your valuable comment.
- This sentence has been modified as follows:
“Recently, the TC21 titanium alloy (Ti-6Al-2Sn-2Zr-3Mo-1Cr-2Nb-Si, wt.%) is considered as a new version in aerospace applications that replaced the commercial Ti6Al4V grade five titanium alloys. It belongs to the family of (α+β) titanium alloys with a high strength, toughness, and damage-tolerance properties [8].”
- Page 1, line 42 - there is “In case of”, but should be ”In the case of”.
- Thanks for your valuable comment.
- This sentence has been modified according to the English editing service as follows:
“The use of TC21 alloy in aircraft components such as landing gear or flap track, there will be a significant advantage on the weight reduction for such aircraft [9].”
- Page 1, line 44 - there is “aircrafts”, but should be ” aircraft”.
- Thanks for your valuable comment.
- This sentence has been modified as follows:
“there will be a significant advantage on the weight reduction for such aircraft [9].”
- Page 2, line 51 - there is “[11].Laser”, but should be ” [11]. Laser”.
- Thanks for your valuable comment.
- This sentence has been modified as follows:
“[11]. Laser processing, the newly developed techniques”
- Page 3, lines 111, 112 - the number and the unit should be written in one line.
- Thanks for your valuable comment.
- This sentence has been modified as follows:
“In this study, TC21 titanium alloy in a cylindrical form of 120 mm diameter and 190 mm length”
- Page 4, line 141 - there is “Table 4. conditions of deposited powder on TC21 substrates.”, but should be ” Table 4. Conditions of deposited powder on TC21 substrates.”.
- Thanks for your valuable comment.
- This caption has been modified according to the English editing service as follows:
“Table 4. Data of deposited powder on TC21 substrates.”
- The numbering of the figures is incorrect in the text. For example, figure 2 is on the page 5 and 7.
- Thanks for your constructive comment.
- It has been modified and all serial numbers of all Figures have been checked.
- Lines 225, 278 - poorly visible markers in figures a, b, c. This should be corrected.
- Thanks for your constructive comment.
- Visible markers in figures a, b, c have been added as follows:.
|
|
(b) |
(c) |
Figure 1. EDAX point analysis results of deposited layer made with 100% Stellite-6 powder (a) gray color, (b) dark gray color, (c) white color and (d) the chemical analysis of each color.
- Lines 237,291,315 - the font size in the figure should be unicolative.
- Thanks for your constructive comment.
- All XRD figures have been replaced by more clear and visible ones.
- Line 278 - in figure a, the x and y axes should be described.
- Thanks for your constructive comment.
- The X and Y axes of Figure 11(a) have been described.
- Line 311- the figure should be read in its entirety on the second page. This should be corrected.
- Thanks for your constructive comment.
- This Figure has been moved for the second page.
- Line 339 – 340 - the number should be written together with the unit on one line.
- Thanks for your constructive comment.
- The number has been moved together with the unit on one line
- Line 374- there is” ()”, ????.
- This part has been removed
- Lines 390-391 - it should be written on one line “process 390 [33]”.
- Thanks for your constructive comment.
- The sentence with the reference has been moved in the same line.
- Line 410 - Delete the border from the figure.
- Thanks for your constructive comment.
- The border has been removed.
- Line 412 - In Figure 11b and 11d a faint marker. This should be reinforced.
- Thanks for your constructive comment.
- Visible markers in figure 11 b, c & d have been added.
- Line 457 - a year is missing.
- Thanks for your constructive comment.
- This reference has been checked and all references as well.
Substantive:
- The article lists some minor linguistic errors on the first page of the article, but it is recommended to correct it throughout the text.
- Thanks for your constructive comment.
- Whole manuscript has been carefully checked by a native English-speaking colleague.
- When there is a reference to a figure in a sub-point in the text, the drawing should be in the same paragraph and not elsewhere (for example: line 180 – Fig. 3. Page 5, line 203, page 6 - another paragraph).
- Thanks for your constructive comment.
- After the revision and the modification of our manuscript we will do it throghtout the whole manuscript.
- The numbering of figures and their references should be confirmed throughout the text. For example, there is no figure 7.
- Thanks for your constructive comment.
- It has been modified and all serial numbers of all Figures have been checked.
- Apart from the SEM and XRD studies, there is no in-depth analysis using TEM.
- Thanks for your constructive comment.
- Extra investigation via transmission electron microscopy (TEM) is planned for future work.
- No statistics (e.g. standard deviation) compiled for the measurements taken on the tested samples. The statistics for the following investigations: dry sliding pin-on-ring wear test, microhardness test should be complied in paper.
- Thanks for your constructive comment.
- An error bars have been attached to the investigation of dry sliding pin-on-ring wear test, and microhardness test, as every experimental test was repeated three times to ensure the validity of the obtained results.
- Therefore, I recommend that this manuscript consider publication, after taking into account the editing and substantive corrections.
Hoping that the changes introduced improved the manuscript in satisfactory way. With our best regards
